# Anaerobic Digestion of Olive Mill Wastewater and Process Derivatives—Biomethane Potential, Operation of a Continuous Fixed Bed Digester, and Germination Index

Jonas Pluschke [1],*, Katharina Faßlrinner [1], Fatma Hadrich [2], Slim Loukil [2], Mohamed Chamkha [2], Sven-Uwe Geißen [1] and Sami Sayadi [3]

1   Environmental Process Engineering, Technische Universität Berlin (TUB), 10623 Berlin, Germany; k.fasslrinner@campus.tu-berlin.de (K.F.)
2   Laboratory of Environmental Bioprocesses, Centre of Biotechnology of Sfax, P.O. Box 1177, Sfax 3018, Tunisia; mohamed.chamkha@cbs.rnrt.tn (M.C.)
3   Biotechnology Program, Center for Sustainable Development, College of Arts and Sciences, Qatar University, Doha 2713, Qatar
*   Correspondence: jonas.pluschke@tu-berlin.de

**Abstract:** Olive mill wastewater (OMW) management is an economic and environmental challenge for olive oil-producing countries. The recovery of components with high added value, such as antioxidants, is a highly researched approach that could help refinance performant wastewater treatment systems. Anaerobic (co-)digestion is a suitable process to valorize the energetic and nutritional content of OMW and OMW-derived waste streams from resource recovery processes. Issues of process stability, operation, and yields discourage industrial application. Deepening the understanding of biomethane potential, continuous anaerobic digester operational parameters, and co-substrates is key to large-scale implementation. The biomethane potential of different OMW-derived samples and organic solid market waste as co-substrate was 106–350 NL methane per kg volatile solids (VS). The highest yields were obtained with the co-substrate and depolyphenolized OMW mixed with retentate from an ultrafiltration pretreatment. Over 150 days, an anaerobic fixed-bed 300 L digester was operated with different OMW-derived substrates, including OMW with selectively reduced polyphenol concentrations. Different combinations of organic loading rate and hydraulic retention time were set. The biogas yields ranged from 0.97 to 0.99 L of biogas per g of volatile solids (VS) eliminated, with an average methane content in the produced biogas of 64%. Potential inhibition of the process due to high polyphenol concentrations or over-acidification through volatile fatty acids was avoided in the continuous process through process and substrate manipulation. High concentrations of potassium and low concentrations of nitrogen and phosphate end up in the digestate. Sulfate reduction results in high $H_2S$ concentrations in the biogas. The digestate was tested for phytotoxic properties via the germination index. Diluted digestate samples improved germination by up to 50%.

**Keywords:** olive mill wastewater; biomethane potential; fixed bed biogas digester polyphenols; germination index

## 1. Introduction

Olive mill wastewater (OMW) is a highly relevant agricultural waste stream in the Mediterranean area. More than 30 million m³ arise each season worldwide. Tunisia has a share of about 8% [1]. The challenges related to the management of OMW were one of the drivers for the conversion of olive mills to more efficient 2-phase decanters that separate the oil from a wet pomace instead of OMW and dry solids. However, millions of cubic meters of OMW arise every season in olive oil-producing countries. They are responsible

for vast amounts of GHG emissions, contamination of surface and groundwater, and smelly nuances [2–4].

The anaerobic (co-)digestion of OMW has been researched since the 1980s, however with limited industrial applications [5–7]. Much like in the case of cow and swine manure in Central Europe, anaerobic digestion is a suitable valorization path for this important waste stream. The technology is fully developed and optimized on an industrial scale.

There are several reasons why, with the exception of a few large-scale plants in Italy, Spain, and Greece, there is no large-scale implementation of anaerobic technology for the valorization of OMW as the main or co-substrate in agricultural biogas digesters: (1) The seasonal and uneven occurrence of OMW over the year; (2) the weather dependent properties of the substrate; (3) the potential inhibition of the biodegradation by high molecular weight polyphenols or potassium and with its high organic load (COD can exceed 100 $kg \cdot m^{-3}$); and (4) the risk of over-acidification due to the low initial pH. The general feasibility of the process is well-researched and documented. To be a viable substrate and energy source, the challenges of process stability, yield, co-fermentation, and digestate utilization need further investigation [8]. Only an in-depth understanding of inhibition effects, optimal operating parameters, maximum biogas potentials, and holistic digestate concepts can give plant operators and planners the necessary certainty that large-scale plants will continue to operate stably when OMW is added.

OMW carries enormous potential regarding nutrients, water, and energetic valorization. Cow slurry is now an incremental part of the nutrient supply for agriculture in Germany. Around 3000 biogas plants produced 4.0 TWh of electricity in 2016 [9]. A similar scenario with OMW in the Mediterranean is conceivable if the process control and inhibition issues are well understood and managed.

The effect of a newly developed polyphenol extraction process for selective removal of low molecular weight polyphenols inhibiting anaerobic degradation was investigated in biomethane potential batches and an anaerobic continuous fixed bed digester. All OMW-derived waste streams of the process (sediment, ultrafiltration retentate, swim layer, flushing water, and depolyphenolized OMW), as well as a slurry of organic market waste, were investigated. The phytotoxic properties of the resulting digestate were quantified using the germination index.

## 2. Materials and Methods

**Olive mill wastewater:** The olive mill wastewater samples were all derived from four different three-phase or traditional mills in Sfax, Tunisia, in the season '22/'23. OMW 1 was collected from the same olive mill on two different dates within one month. Numerous olive kinds were mixed from many groves, but the most common is Chemlali [10]. The samples for analysis were taken and stored at 4 °C. The substrates for feeding the biogas reactor were stored outdoors under the sun and rain shelter for several weeks, depending on the feeding schedule of the biogas reactor.

Depolyphenolized OMW samples were subjected to mechanical pretreatment, ultrafiltration, and a selective polyphenol extraction process by adsorption. The extracted polyphenols were further purified for valorization in industrial applications, such as feed additives or cosmetics [11]. The residue from the ultrafiltration process was reduced to roughly 40% of the feed volume. All OMW (derived) samples used for the experiments are characterized in Table 1.

**Table 1.** Composition and timetable of the substrates of the anaerobic bioreactor.

| OMW | Experiment [a] | Total Polyphenols [b]/Hydroxytyrosol [c] | TOC/DOC | TS/VS | COD | $SO_4^{2-}$ | $K^+$ | $PO_4^{-2}$ | Total Nitrogen | Treatment and Mixture |
|---|---|---|---|---|---|---|---|---|---|---|
| Unit | Experiment (Days) | $g \cdot L^{-1}$ | $g \cdot L^{-1}$ | $g \cdot L^{-1}$ | $g \cdot L^{-1}$ | $g \cdot L^{-1}$ | $g \cdot L^{-1}$ | $g \cdot L^{-1}$ | $g \cdot L^{-1}$ | - |
| OMW 1 raw | BMP, GI, BOD | 3.36/- | 37.6/24.0 | 74.1/60.6 | 90.3 | - | - | 0.45 | 0.60 | OMW 1 Permeate after adsorption (to recover polyphenols) |
| OMW 1retentate | BMP, BOD | 2.64/- | 59.8/25.0 | 100.5/86.3 | 193.3 | - | - | 0.50 | 1.26 | OMW 1 sieved + filtered with ceramic ultrafiltration membranes |
| OMW 1 treated | I AD: (40), GI, BOD, BMP | 1.4–2.4/0.18–0.54 | 17.9/16.8 | 39.1/27.6 | 49.1 | 1.3–1.7 | 4.8 | 0.31–0.41 | 0.18 | 100% 3-phase OMW after membrane filtration and adsorption |
| OMW 2_1 diluted | II AD: (16) | 1.32/- | 12.1/10.9 | 25.0/19.6 | 40.0 | 0.8 | 2.7 | 0.15 | 0.11 | 43% sieved OMW 2 (traditional method), 57% water |
| OMW 2_2 diluted | III AD: (22) | 1.66 | 15.2/13.6 | 31.4/24.6 | 50.2 | 1.0 | 3.4 | 0.19 | 0.14 | 53% sieved OMW 2, 47% water |
| OMW 3treated | V AD: (43) | 1.56/0.57 | 18.8/11.9 | 39.5/29.0 | 56.6 | 2.1 | 3.3 | 0.27 | 0.37 | 75% 3-phase OMW after membrane filtration and adsorption, 25% retentate from the membrane filtration treating OMW |
| OMW 4treated | VI AD: (26) | 1.56/0.98 | 12.2/11.5 | 24.5/17.7 | 33.9 | 0.4 | 3.0 | 0.28 | 0.16 | |
| Co-Substrate | BMP | - | - | 89.2 [d]/81.8 [d] | 85.0 | - | - | - | - | Solid organic food-market waste |
| Inoculum | BMP | - | 12.37/1.86 | 28.6/17.5 | - | | | 0.01 | 1.76 | Digestate from an anaerobic reactor that is fed with market waste |
| Digestateday 37 | GI | 0.68 | 2.34/2.00 | 22.3/11.4 | 8.79 | 0.19 | 4.79 | 0.08 | 0.14 | Digestate of the fixed bed reactor used for the continuous experiment |

[a] Experiment: BOD: biological oxygen demand; BMP: biomethane potential; AD: anaerobic digester; GI: germination index. [b]: expressed in gallic acid equivalents. [c] Hydroxytyrosol is the strongest antioxidant of polyphenols (generally in nature). [d] g/kg.

**Biomethane potential:** An adapted method based on Hollinger et al. [12] and VDI 4630 (2016) was applied to determine the biomethane potential of the treated and untreated olive mill wastewater. Duplicates of five different samples were measured in 1 L Schott glass bottles, filled 2/3 (666 mL) with an inoculum-substrate ratio of 1:1. The digestate of an anaerobic biodigester treating market waste was used as an inoculum and mixed with three substrates derived from the same OMW sample, as characterized in Table 2. All samples are flushed with nitrogen prior to the experiment, sealed, and incubated at 37 °C. The produced biogas was measured using the liquid replacement method, utilizing NaOH to remove $CO_2$.

**Table 2.** Composition of the BMP samples (all OMW samples derive from OMW 1).

| | 1 Inoculum | | 2 Raw | | 3 Depolyphenolized OMW | | 4 Co-Substrate | | 5 Retentate | |
| | Volume | VS | Volume | VS | Volume | VS | Volume | VS | Volume | VS |
| **Sample Name** | **L** | **g** | **L** | **g** | **L** | **g** | **L** | **g** | **L** | **g** |
| Inoculum | 0.35 | 6.11 | 0.35 | 6.11 | 0.35 | 6.11 | 0.35 | 6.11 | 0.35 | 6.11 |
| OMW | | | 0.10 | 6.11 | | | 0.07 | 4.28 | | |
| Depolyphenolized OMW | | | | | 0.21 | 6.11 | | | 0.10 | 3.06 |
| Co-Substrate | | | | | | | 0.02 | 1.83 | | |
| Retentate | | | | | | | | | 0.04 | 3.06 |
| Water | 0.32 | | 0.22 | | 0.11 | | 0.22 | | 0.18 | |
| Total | 0.67 | 6.11 | 0.67 | 12.23 | 0.67 | 12.23 | 0.67 | 12.23 | 0.67 | 12.23 |

**Anaerobic digester:** The digester used for the upscaled experiments [13] is a 1.95 m high, double-jacketed, thermostated stainless-steel column with an internal diameter of 0.45 m, resulting in an active fixed bed volume of around 270 L. It is filled with about 70 kg of carriers for biomass retention (Hiflow PVC rings—size 38-1), resulting in a specific surface of around 200 $m^2 \cdot m^{-3}$ and a gap degree of over 92%. The temperature was kept at 37 °C with a thermostat. The feed was dosed with a BG 600 FJ-S peristaltic pump at a flow rate of 0.7 $L \cdot min^{-1}$ (November–January) and 0.8 $L \cdot min^{-1}$ (February). The inflow pH was adjusted with a 5 mol $L^{-1}$-NaOH solution to pH 6. Biogas and digestate samples were taken irregularly to determine the methane content by gas chromatography. The feed substrates and durations are depicted in Table 1. The effects of operating parameters such as hydraulic retention times and organic loading rates on degradation efficiency and biomethane production were observed and determined over 150 days.

**Germination Index:** The method is based on a modified Zucconi test and DIN EN 16086-2: Soil improvers and growing media—Determination of plant response—Part 2. Triplicates of 20 seeds are placed into Petri dishes with filter paper in the dish and in the lid. Lettuce Romaine seeds are used (Variety: NADER, Brand: Enza Zaden, Batch No. 6.121.702, Germination and Purity: 99%, Origin: South Africa). A total of 5 mL of sample is added per Petri dish, and 1 mL of distilled water is put on the filter paper in the lid of the Petri dish. For the control, 5 mL of distilled water was added to the seeds instead of the sample. The closed Petri dish is kept in a dark place at room temperature until day 5. Every day, the number of germinated seeds is counted. On the fifth day, the root length of all germinated seeds was measured. The Germination Index is calculated with the Formula (1):

$$GI\ (\%) = (NG_S \cdot ARL_S \cdot NG_C^{-1} \times ARL_C^{-1}) \cdot 100 \tag{1}$$

GI: germination index; NG: number of germinated seeds; ARL: average root length; S refers to the sample; and C refers to the control.

**Analytical methods:** The Analytik Jena TOC analyzer multi N/C 3100 (Jena, Germany) was used to determine total organic carbon (TOC), dissolved organic carbon (DOC), and total nitrogen (TN). For DOC measurements, the sample was filtered with a Whatman 0.45 μm membrane filter (Kent, UK). The chemical oxygen demand (COD) was measured

based on the thermal disintegration of the samples at 1200 °C using the QuickCODlab-03D0318 analyzer and autosampler from LAR Process Analysers AG. The pH and conductivity were measured using an OHAUS Starter 2100 and a Consort C651 sensor. The Folin–Ciocalteau method was used to determine the polyphenol concentration in gallic acid equivalents. The following protocol, which was optimized at the Technical University of Berlin (TUB), was used to prepare 96-well plates: (1) add 90 μL ultrapure water; (2) add 20 μL of gallic acid standard or sample diluted to the calibrated concentration; (3) add 10 μL Folin–Ciocalteau reagent; and (4) add 100 μL $Na_2CO_3$ (10.75%). The reaction time is 30 min in the dark (first 15 min shaking at 100 rpm). The light absorption at 620 nm is measured. The blank sample was subtracted, and gallic acid equivalents were calculated based on a linear calibration curve (gallic acid: 800, 400, 200, 100, 50, and 25 mg·$L^{-1}$). The dry matter (DM) was determined in accordance with DIN 12,880 by drying the sample for 24 h and weighing the mass difference. The dried samples were oxidized at 600 °C for 6 h to remove all organic components and re-weighted to determine the organic dry matter (VS). To quantify the methane content in the biogas, the sample vials are flushed for 5 min with nitrogen gas. The vial is then flushed with the produced biogas for at least 5 min and afterward analyzed in a gas chromatography system (Agilent Technologies 7890A GC System with a Bora bond q column and Agilent Technologies G188, Network Headspace Sampler). Based on the Nordmann method and a titration test adapted by the German Federal Agricultural Research Center (FAL), the ratio between the volatile fatty acids and the total alkalic carbonates (FOS/TAC) was measured. A total of 20 mL of sample is placed on a magnetic stirrer and homogenized consistently. Using the SCHOTT TitroLine Easy, 0.1 N $H_2SO_4$ (=0.05 mol·$L^{-1}$) was titrated until the sample reached pH 5. The time and amount of acid needed are recorded. The titration is then continued until pH 4.4 is reached, and the needed time and amount of acid are recorded again. FOS and TAC are then calculated with Formulas (2) and (3):

$$TAC = H_2SO_4\text{-Volume added from start to pH 5 in mL} \times 250 \qquad (2)$$

$$FOS = (H_2SO_4\text{-Volume added from pH 5 to pH 4.4 in mL} \times 1.66 - 0.15) \times 50 \qquad (3)$$

Ion concentrations were determined via ion chromatography by Metrohm using a Metrosep A Supp 17—150/4.0 column for anions and a Metrosep C 4—150/4.0 column for cations. All samples were filtered and diluted to the calibrated concentrations of 20–100 ppm. The determination of the biological oxygen demand 30 ($BOD_{30}$) was carried out in accordance with DIN EN 1899-2 using Oxi Top Control of WTW for four samples, all deriving from different treating steps of the OMW. The test was carried out in triplicate with two different dilutions for each sample. The incubation period was 30 days at a temperature of 20 °C.

## 3. Results

### 3.1. Biological Oxygen Demand

For the substrates used in the biomethane potential experiments, the biological oxygen demand was determined to determine the easily and slowly degradable components (Figure 1). For all substrates except digestate, an increase in oxygen demand is observed after day 2, indicating that most readily degradable organics are degraded. At the end of the measurements, after 30 days, the raw OMW and the retentate curves were still slightly increasing, indicating that further degradation of poorly degradable organics was continuing. Membrane filtration and adsorption improve the $BOD_5$ to COD ratio from 1:3.2 (raw) to 1:2.2 (depolyphenolized OMW). In terms of slow-degrading substrates ($BOD_{30}$:COD ratio), the ratio for raw OMW was 1:1.6, and for depolyphenolized OMW, it was 1:1.3. This indicated slow degradation kinetics for many organics in the wastewater. The retentate has a $BOD_5$ to COD ratio of 1:6, with a $BOD_{30}$ to COD ratio of 1:2, which leads to the conclusion that long retention times or dilution are necessary for the degradation.

As for the digestate, the $BOD_{30}$ to COD ratio is 1:2, indicating that the remaining organics require more adapted biomass and longer retention times for further degradation [14].

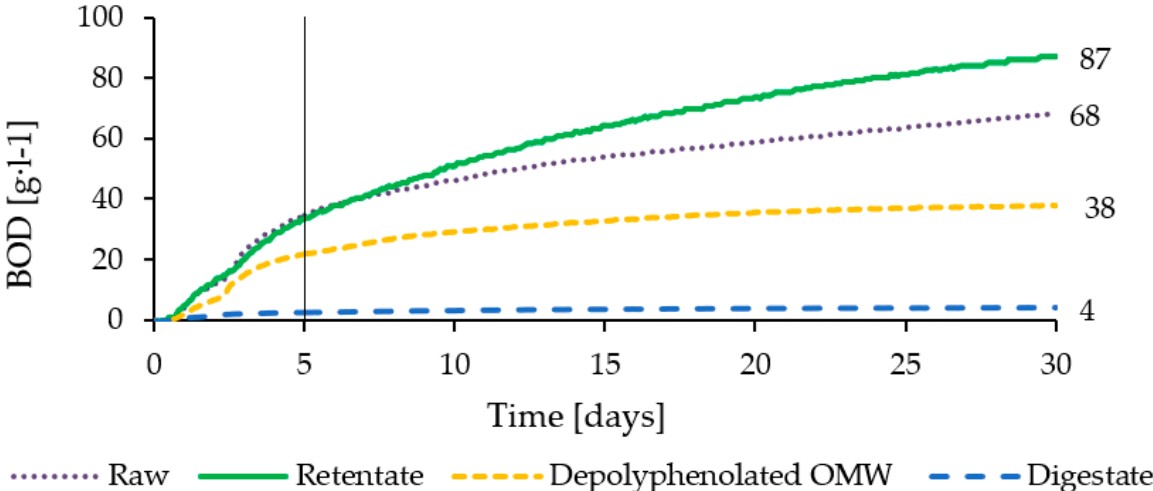

**Figure 1.** Biological oxygen demand over 30 days of OMW-derived substrates.

*3.2. Biomethane Potential*

The BMP batches were incubated for 105 days, showing the generally slow kinetics of the anaerobic degradation. The biogas yield is shown in Figure 2, compared to the inoculum baseline. Due to different technical issues, three samples were not quantified. The baseline of the graph is the methane yield of the inoculum at $731 \pm 60$ mL after 105 days. The variation in the inoculum BMP results in a margin of error of approximately 16%, which affects all results in addition to the variations in the BMP samples. Due to differences in the obtained yields, both batches containing co-substrate are depicted separately. The highest yields were achieved by mixing the OMW with a slurry of organic market waste and by depolyphenolized OMW mixed with retentate, both reaching 350 NL methane per kg of organic dry matter. The volatile solids content was reduced by 6.4 g and 7.8 g VS, respectively. Initial inhibition was observed for depolyphenolized OMW and raw OMW with co-substrate, with lower biomethane production rates than the inoculum for 20 and 65 days, respectively. This is unexpected and likely due to the higher-than-expected and fluctuating biogas production of the inoculum, combined with the error margin of the measurement method. Raw OMW, raw OMW with co-substrate, and depolyphenolyzed OMW with retentate showed retarded biogas production, with a second steep increase in methane production between days 30 and 45. This is likely linked to an adapted microbiome in the samples after extensive incubation [15]. Using digestate from an anaerobic digester that is fed with olive mill wastewater is an option to reduce this lag phase. Micoli et al. indicated that adding biochar is a viable option to stabilize the incubation phase [16]. By reducing the effect of the degradation of the inoculum on the overall result, increasing the VS ratio between the inoculum and substrate can increase the accuracy of the experiment.

On day 6, the incubation temperature for all samples was 8 degrees higher than during the rest of the period due to a technical problem. It is noticeable that on this day, there is a decrease in methane production for most substrates, which is likely connected to the higher pressure in the vessel due to the higher temperature. The gas yield was adapted to normal conditions (273 K and 1.013 bar).

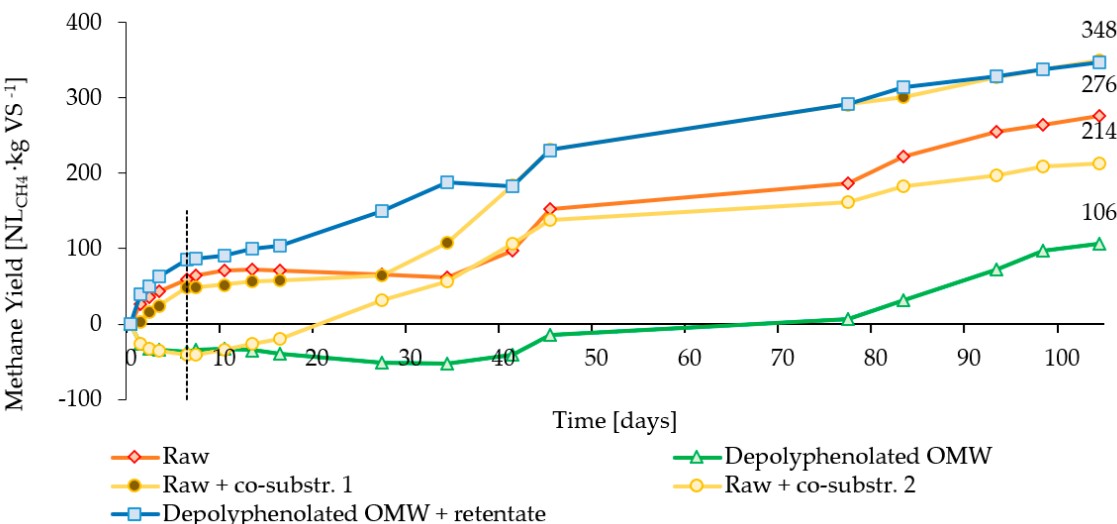

**Figure 2.** Biomethane potential with the inoculum as a baseline.

### 3.3. Continuous Anaerobic Digestion of Raw and Derived OMW

The fixed-bed biogas reactor was operated over 150 days, utilizing different OMW-derived substrates (Table 1). To investigate the effects on the control of process stability and degradation efficiency, the organic loading rates (OLR) and hydraulic retention times (HRT) were varied (Figure 3). The OLR varied between 1.8 and 5.8 kg COD·m$^{-3}$·d$^{-1}$ and the HRT between 10 and 27 days. (Figure 3).

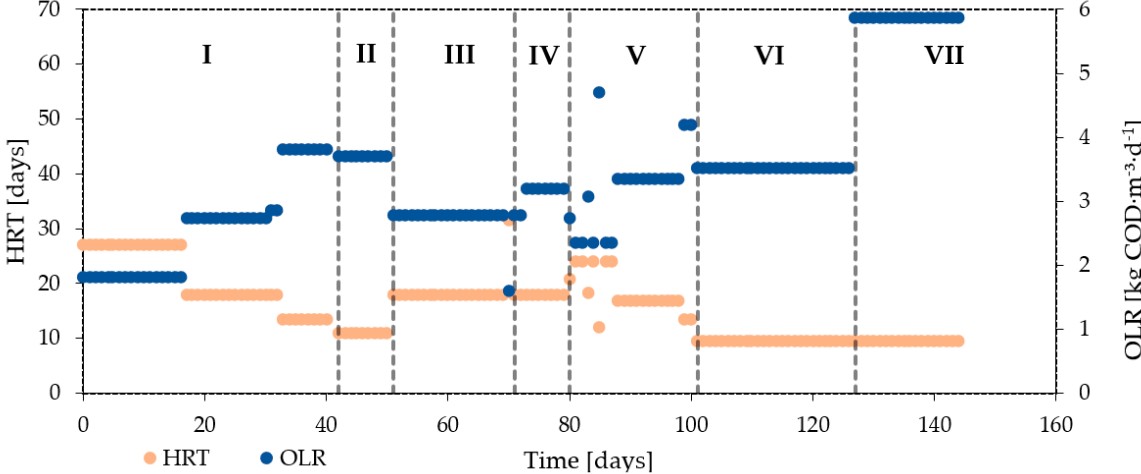

**Figure 3.** Chronological overview of OLR, HRT, and used substrates time periods of the substrates: I: OMW 1, II: OMW 2_1, III: OMW 2_2, IV: OMW 2_3, V + VI + VII: OMW 3.

Three phases can be distinguished in the substrate: The first phase lasted 40 days, and exclusively depolyphenolized OMW was fed. In those 40 days, the organic loading rate doubled from 2 to 4 kg COD·m$^{-3}$·d$^{-1}$ in two steps after 2 weeks of stable operation. From days 41 to 78, diluted OMW was used, with an average OLR of 3.1 kg COD m$^{-3}$ d$^{-1}$. Tap water was added to decrease the viscosity and concentration of potentially inhibiting monomeric polyphenols. The third and longest phase lasted from days 79 to 148 (phases V–VII, with substrates derived from OMW3 in the depolyphenolization process). The mixing ratio of the different OMW substrates is the same as that produced during polyphenol extraction (25–45% retentate and 55–75% depolyphenolized OMW).

Figure 4 shows the daily biogas production and the measured methane content. The horizontal lines indicate the average biogas production for different OLRs, displaying a clear correlation. An increase in the produced gas volume is visible with increasing

OLR. The performance of the reactor was stable between days 77 and 86, but the biogas production was not recorded due to a technical issue with the biogas meter. The measured methane content varies between 51 and 78%, with an average of 64%. The obtained yields vary between 0.97 and 0.99 L biogas·(g $VS_{elim.}$)$^{-1}$ or 0.44–0.5 L biogas·(g $COD_{elim.}$)$^{-1}$ (0.4–0.46 L biogas ·(g $COD_{elim.}$)$^{-1}$ at standard conditions). The methane yield is 0.29–0.33 L $CH_4$·(g $COD_{elim.}$)$^{-1}$ (0.26–0.30 L $CH_4$·(g $COD_{elim.}$)$^{-1}$ at standard conditions). This corresponds to 76–86% of the theoretical methane maximum that can be obtained (0.38 L·g COD at 25 °C [17]). The yields for the period with the inflow derived from OMW 3 (0.33–0.34 L $CH_4$·(g $COD_{elim.}$)$^{-1}$) cannot be considered due to the inhomogeneous substrate. In the first days, the reactor effluent still contained digestate from the previous influent, which had lower COD and VS values. This leads to an overestimation of the yield obtained in the first days, which is not considered in the evaluation.

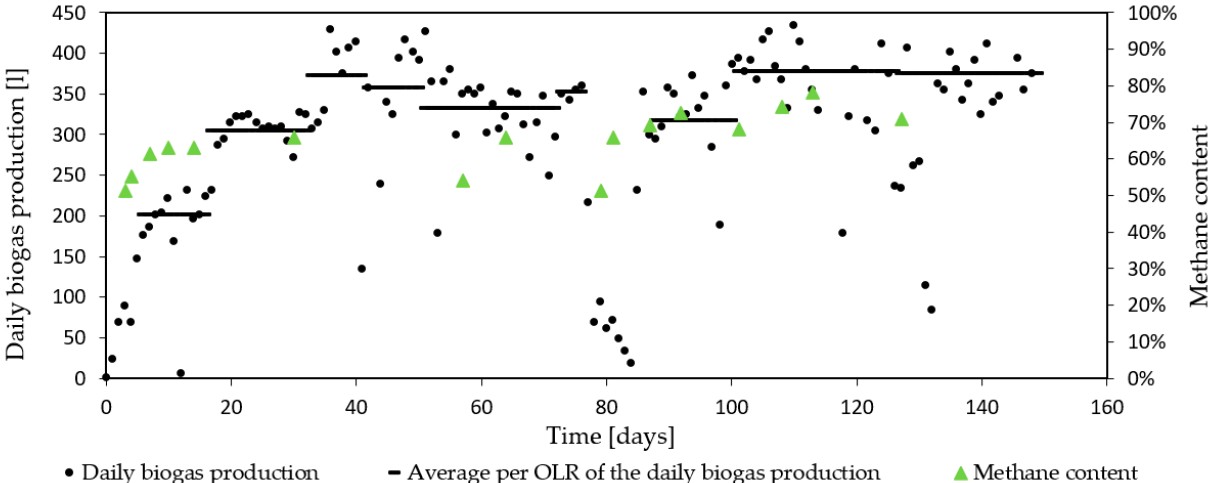

**Figure 4.** Daily biogas production, methane content, and average daily biogas production for each OLR phase are depicted in Figure 3.

The FOS/TAC ratio is a practical indicator that can be determined with simple laboratory equipment and is used by many operators of biogas digesters to determine the biochemical state of the digesters. The optimum is between 0.3 and 0.4. The pH of the substrate was adjusted to 6 for the first 70 days of the experiment using a 20% NaOH solution (5 mol L$^{-1}$) to prevent over-acidification. After the pH adjustment of the feedstock was terminated, the FOS/TAC increased. The data were generated only until day 96 of the experiment, with a value of 0.38, slightly over the optimum. The microbiome was stable and able to treat the acidic OMW without pH adjustment, producing a digestate with a pH between 7.0 and 7.8 (Figure 5). Lower HRT and higher OLR resulted in higher hydrolysis and acidification. The stable pH at the outflow indicates that higher OLRs are likely feasible without overacidification [18,19].

Figure 6 shows the VS feed and outflow concentrations together with the resulting degradation efficiencies. The average removal of VS was 71%. A set of outliers with degradation rates below 60% are partly due to hardware issues and are also due to imperfect sampling and analytical errors. Generally, a decrease in reaction velocity is possible due to the inhibitory character of raw OMW that was fed in the first 41 days. The DM/VS ratio of 57% in the digestate indicates advanced mineralization of organic carbon. The ratio TOC:VS:COD of the feed averaged 2:3:6. After the anaerobic digester, this changed to 1:3:4, indicating that the COD is only partially related to organic carbon and likely to the non-carbon portion of the VS.

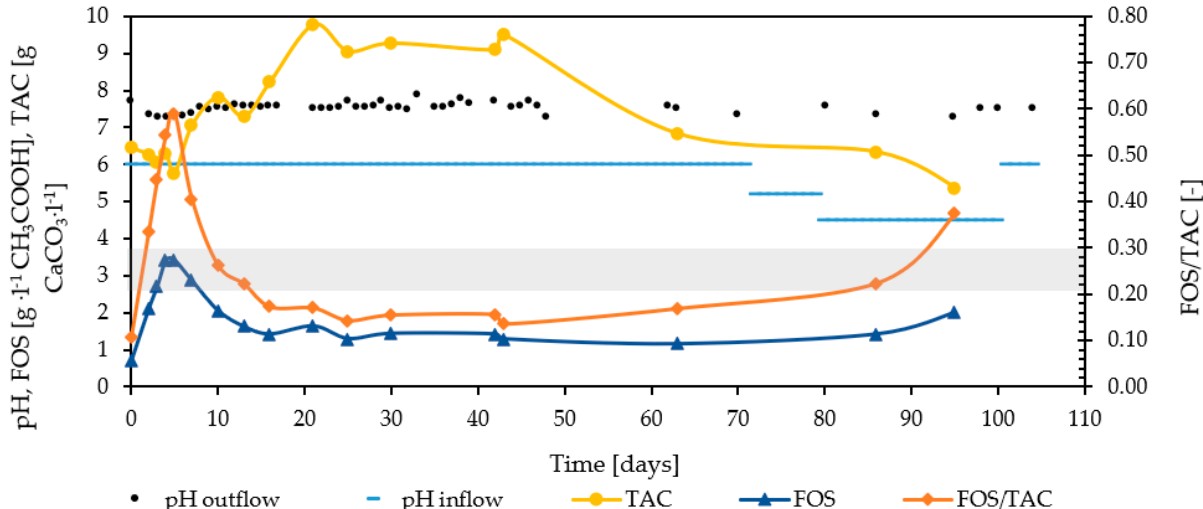

**Figure 5.** FOS, TAC, FOS/TAC, and pH feed and outflow. The grey bar indicates an ideal FOS/TAC value between 0.2 and 0.3.

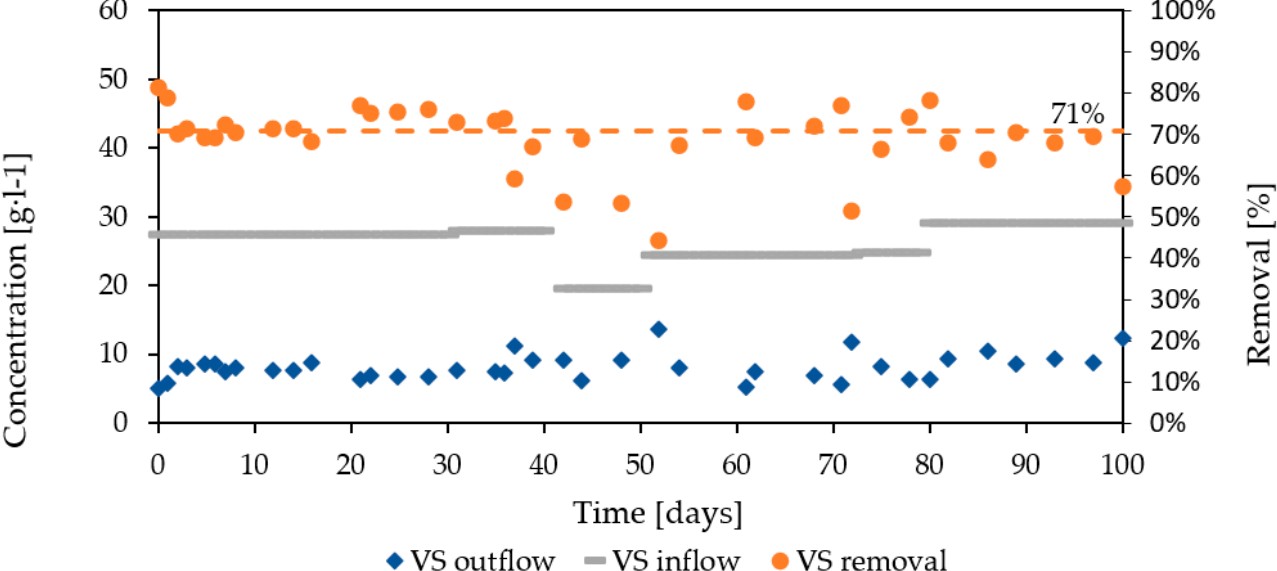

**Figure 6.** VS concentrations of the inflow and outflow with the resulting degradation efficiencies. The dotted line represents the average VS removal rate over the course of 100 days.

With the TOC to COD ratio, the oxidation state of the inflow can be calculated, which allows an indication of its main components and an estimation of the methane content in the produced biogas [20,21]. The inflow derived from OMW 1 and OMW 4 has a COD to TOC ratio of 2.7, leading to an oxidation state of $-0.12$ and $-0.17$ [20]. The COD to TOC ratio for OMW 2 is 3.31 and 3.01 for OMW 3, which corresponds to an oxidation state of $-0.52$ and $-0.96$. A lower oxidation state leads to a higher methane content in the produced biogas [21], which is consistent with the measured increase in methane content (68–78%) during the feed period of OMW 3. Calculating the estimated methane content proposed by Henze et al. [21] % $CH_4$ = COD/TOC·18.75, the estimated methane content is 52–62%, which is 10% lower than the measured methane values for the corresponding period.

The feed COD concentration ranged from 35 to 58 g·L$^{-1}$ in the first 105 days of operation of the continuous anaerobic digester. The COD degradation averaged 71%, with a residual COD of 11 g L$^{-1}$. Most of this COD in the digestate is not caused by organic carbon, as the TOC:COD ratio is low at 1:4. The biological oxygen demand was

not determined for this sample. However, it is likely that the digestate is stabilized and suitable for soil application.

The inflow has a low nitrogen content, with an average ratio of COD:TN:P of 170:0.7:1, typical for OMW. Considering the ideal ratio of 300:5:1 for the anaerobic degradation of carbohydrates, the carbon-to-nitrogen ratio is not optimal. A mixture of the substrate or digestate with nitrogen- and phosphate-rich biomass, like manure, will provide sufficient nutrients.

### 3.4. Germination Index

The results of the germination tests in Figure 7 showed a decrease in phytotoxicity with increasing dilution. Treatment of OMW by membrane filtration and adsorption, particularly the subsequent anaerobic digestion, reduces phytotoxicity. However, undiluted digestate showed high phytotoxicity with a GI of 6%. The addition of 1 and 6.5% digestate to distilled water improved germination by 50 and 23%, respectively. Figure 7 depicts the results of the germination test for the digestate after 5 days. After germination, there was an increase in the growth rate of the seedlings and roots, but this was not quantified. Further research is needed on the fertilizing effect on adult plants and trees.

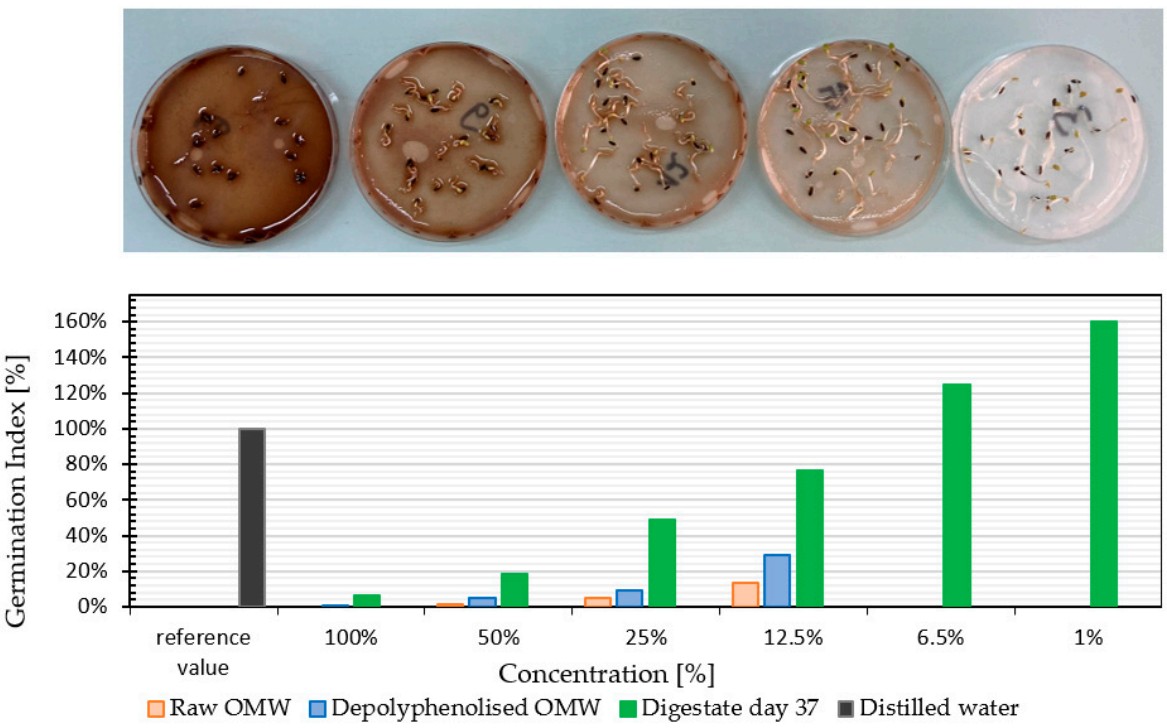

**Figure 7.** The picture depicts the digestate sample (day 37) in different dilutions (100%, 50%, 25%, and 13%) and distilled water on the 5th day of the GI experiment. The bar graph depicts the percentage of germinated seeds compared to distilled water, clearly indicating positive effects for highly diluted digestate. The fertilizing effects cannot be quantified by this experimental setup.

## 4. Discussion

The BMP value for untreated OMW of 276 NL CH$_4$· (kg VS)$^{-1}$ is comparable to the result of Calabro et al. [15], who obtained in a mesophilic BMP-test of raw OMW a methane yield of 243 NL CH$_4$·(kg VS)$^{-1}$ at a total polyphenol concentration of 1 g·L$^{-1}$.

The removal of polyphenols did not show a positive effect on the methane yield, as the methane yield of the OMW after adsorption is 62% lower in comparison to the untreated OMW. Besides the polyphenol content, the main difference is the amount of solids. Blika et al. [15] studied the effect of pretreatment steps on the anaerobic digestion of OMW in a mesophilic 3-L CSTR. The authors emphasized the presence of solids as a crucial factor for process stability. After a thermal pretreatment step and sedimentation to

remove solids, stability and methane production rates decreased compared to the anaerobic digestion of raw OMW.

Afif and Pfeifer [22] investigated the biomethane potential of 3-phase olive mill solid waste under mesophilic and thermophilic conditions. The result for the cumulative methane yield obtained after 60 days of digestion at mesophilic conditions was 139 NL $CH_4 \cdot (kg\ VS)^{-1}$. For thermophilic conditions at 55 °C, a higher methane yield of 239 NL $CH_4 \cdot (kg\ VS)^{-1}$ was achieved.

Karray et al. [13] used the same fixed bed reactor with diluted OMW mixed with 10% liquid poultry manure and produced a maximum biogas production of 0.507 L $\cdot$ (g CODintroduced)$^{-1}$.

Numerous approaches to improving process stability and biogas yields have been investigated. Micoli et al. [16] have found a positive effect of biochar addition to the digestion mixture regarding methanogenesis inhibition and digestate phytotoxicity. Gonçalves et al. [23] have investigated an anaerobic hybrid reactor to treat a mixture of piggery effluent and OMW. The reactor proved stable, with biogas production rates of 3.16 $m^3\ m^{-3}\ d^{-1}$ at an organic loading rate of 7.1 kg $m^{-3}\ d^{-1}$. Partially substituting the feedstock of an anaerobic digester for OMW in a relatively short period of time is a key challenge to the treatment of the seasonally arising OMW. Azbar et al. investigated cheese whey and laying hen litter as co-substrates for the anaerobic digestion of OMW, resulting in a strong increase in BMP [24]. Aerobic pretreatment, partial chemical oxidation, flocculation, ultrasound, and others have been investigated to increase process stability [7,25–27]. Bioaugmentation and co-digestion optimization in the mainstream are two options to increase the biomethane yield. Bioaugmentation can be an economical option when dealing with recalcitrant feedstocks or disturbances to inhibitory compounds [28].

The sulfate concentrations between 0.4 and 2.2 $g \cdot L^{-1}$ in the OMW are reduced to $H_2S$ in the biogas, demanding proper desulfurization for biogas valorization.

The high potassium concentration and, to a lesser extent, phosphate content in the digestate make them highly suitable for fertilizing. The concentrations varied between 0.1 and 0.5 g phosphate and 2.7 and 4.8 g potassium per liter of digestate. To produce a full fertilizer, nitrogen-rich substrates like manure should be added.

The increase in the GI with higher dilution is consistent with the observations of Karray et al. [13]. The germination index was determined using cress seeds and a digestate obtained after anaerobic treatment of diluted OMW mixed with 10% liquid poultry manure. The GI was analyzed for the digestate diluted with water at concentrations of 100%, 50%, 20%, and 10%. The undiluted digestate did not show germination, and inhibition of germination was observed at a concentration of 50%. The maximum GI of 154% was obtained at a 20% digestate concentration. Komilis et al. [29] investigated the effects of different pretreatment techniques of OMW on the germination of tomato and chicory seeds using a modified Zucconi test. They concluded that the strongest effect on reducing the phytotoxicity of OMW was the dilution with water, especially at a high dilution ratio of 1:10. Mekki et al. proposed other microbiotests to assess the toxicity of the untreated and treated olive mill wastewaters. They concluded that *V. fischeri* remained the most sensitive strain for monitoring the toxicity of such effluent, which proves its utilization as a standard measure of toxicity again [30]. The fertilizing effects, especially when mixed with nitrogen-rich co-substrates, need to be further investigated and quantified to calculate the economics.

## 5. Conclusions

OMW is a suitable co-substrate in agricultural biogas digesters. As a single substrate, it is inapplicable due to the seasonal character of the biomass stream and the potential inhibition by polyphenols. Selective polyphenol recovery can reverse the inhibitory effects and add further value to the waste stream [31].

The biodegradability of OMW, indicated by the $BOD_5$:COD ratio, can be improved by selectively removing polyphenols. Mixing raw OMW with a co-substrate or mixing the two waste streams of the polyphenol recovery process, depolyphenolized OMW and UF retentate, produces high biomethane potentials of around 350 NmL/g $VS_{eliminated}$ at very

long retention times of over 45 days. The methane content in the produced biogas varies between 61 and 72%. The anaerobic digester was able to continuously degrade an average of 71% with 19 days of HRT. Increasing the HRT from 9 to 29 days did not significantly increase the degradation. The organic loading rate was increased up to 5 kg VS m$^{-3}$ d$^{-1}$ without signs of over-acidification. The FOS/TAC indicated a stable biochemical equilibrium inside the reactor, even after terminating the pH adjustment of the feedstock after 70 days.

The digestate needs to be applied as fertilizer to valorize the very high content of fertilizing minerals and humus buildup. Stabilizing the digestate further in an aerobic process (e.g., composting) is not necessary, as the VS can be reduced by up to 80%. A biogas yield of around 320 NL·(kg VS)$^{-1}$ is sufficient for an economic digester operation. The energetic and nutritional valorization of OMW in biodigesters is a feasible economic approach. Three parameters are decisive for the economic feasibility of biogas production from OMW:

- Biomethane yields are comparable to those of renewable raw materials
- Process stability
- A suitable long-year co-substrate plan with alternative biomass for anaerobic digestion between March and November

Germination tests showed that undiluted digestate and OMW have phytotoxic properties for seeds. However, diluted samples increased germination compared to water. Full-grown plants with established root systems showed significantly increased growth compared to non-fertilized plants.

One policy that helped increase the number of small and medium-sized biogas plants in Germany for local renewable energy production was part of the *Erneuerbaren Energie Gesetz,* which provided a degressive payment for small manure biogas plants up to 100 kW$_{el}$. In this way, a local valorization that helps reduce the negative impact of the waste stream on the climate and environment and generates renewable energy is economically feasible. In addition, regions where OMW management is an issue can implement regulations to incentivize or require biogas plant operators to accept a certain amount of OMW as a co-substrate that is diluted enough to have a minimal impact on biogas yields or process stability.

Anaerobic digestion of OMW as a co-substrate in industrial-scale biogas digesters is the most realistic solution for long-term sustainable OMW management. Further research on co-substrates, process stability, and kinetic optimization can reduce business risks for industrial implementation.

**Author Contributions:** Conceptualization, J.P., K.F., S.S., S.L., F.H. and S.-U.G.; methodology, J.P., K.F., F.H. and S.L.; software, J.P.; validation, J.P. and K.F.; formal analysis, J.P. and K.F.; investigation, J.P. and K.F.; resources, J.P., S.S., S.-U.G., S.L. and M.C.; data curation, J.P., K.F., S.L. and F.H.; writing—original draft preparation, J.P. and K.F.; writing—review and editing, J.P., K.F., S.-U.G. and S.S.; visualization, J.P. and K.F.; supervision, M.C. and S.-U.G.; project administration, J.P. and F.H.; funding acquisition, S.S., S.-U.G. and J.P. All authors have read and agreed to the published version of the manuscript.

**Funding:** The InnoVa research project (2nd German-African Innovation Promotion Prize: Prof. Sami Sayadi, Prof. Sven Geißen) was funded by the *Bundesministerium für Bildung und Forschung,* grant number 01DG20005, and managed by the *Deutsche Luft-und Raumfahrtzentrum—Projektträger.* This research was supported by the Ministry of Higher Education and Scientific Research-Tunisia under a contract program for the Laboratory of Environmental Bioprocesses (LR01CBS2015). We acknowledge support by the German Research Foundation and the Open Access Publication Fund of TU Berlin.

**Institutional Review Board Statement:** Not applicable.

**Informed Consent Statement:** Not applicable.

**Data Availability Statement:** The data presented in this study are available on request from the corresponding author. The data are not publicly available due to third party liability.

**Acknowledgments:** Many thanks to CBS and TUB laboratory and technical personnel.

**Conflicts of Interest:** The authors declare no conflict of interest.

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
