# Peer review of "Anaerobic Digestion of Olive Mill Wastewater and Process Derivatives—Biomethane Potential, Operation of a Continuous Fixed Bed Digester, and Germination Index"

_applsci, doi:10.3390/app13179613_

Round 1
Reviewer 1 Report
This paper mainly explores the potential biogas yields in small batches, real yields in a continuous biogas reactor and investigate the detoxifying effect of anaerobic digestion on OMW via germination index. The article is comprehensive and has significance for publication in this journal. However, the description and general writing in the article remain to be further improved, some of the main content and conclusions of the article need to be talked about in more depth. In all, the manuscript could be accepted after major revision. Some detailed questions and comments are shown below:
1. The abstract is merely a statement of data and does not provide a good overview of the purpose and significance of the paper.
2. Too many keywords, preferably limited to 3-5.
3. Are some of the symbols used inappropriate and need to be checked carefully, for example: in line 116, does 1,95 m mean 195 m or 1.95 m?
4. The format of all figures needs to be carefully checked, and some of the symbols in the charts are incorrectly superscripted and subscripted, for example, figure 1.
5. In line 191-192, the sentence “but there were technical problems with the biogas measurement”. What was the problem? How was it solved? Were the data obtained accurate?
6. The conclusion is too long and should be revised to be written more concisely and with a clearer idea.
7. The cited references are relatively small, and it is recommended to add some references on this research.
No suggestions
Reviewer 2 Report
In this manuscript, OMW was used to produce biogas. Over 150 days an anaerobic fixed bed 300L digester was operated with different OMW derived substrates, including OMW with selectively reduced polyphenol concentrations. The manuscript is well written and well organized. However some comments should be considered before any further recommendations:
1- OMW contains many toxic but valuable materials including biophenols. The more economic approach is to recover such compounds then biological treatment processes can be more successfully conducted. This fact should be mentioned in the introduction.
2- The recovery of the valuable contents of OMW prior the treatment is a new approach in sustainability. Please see the following review and similar many others. this approach could be formulated as one of the recommendations for future work for researchers working in this and similar fields>
a- Zakaria Al-Qodah, Habis Al-Zoubi, Banan Hudaib, Waid Omar, Maede Soleimani, Saeid Abu-Romman, Zacharias Frontistis Sustainable vs. Conventional Approach for Olive Oil Wastewater Management: A Review of the State of the Art, Water 14 (11), 1695, 2022.
3- Some recommendations for future work can be added after the conclusion section.
None
Reviewer 3 Report
The manuscript is very hard to fallow. There are grammar and English mistakes. There are no consistency in terms such as sample, substrate, treatment etc.. The goals and motivation of the study are not well presented. The experiments are not well described. It is unclear how the experiments serve the goals of the study
There are grammar and English mistakes
Round 2
Reviewer 3 Report
There are some improvements in the manuscript, but it still very hard to fallow. The background is poor. The introduction should be improved. The experiment method is very hard to understand. The goals and motivation of the study are not well presented. Still, it is unclear how the experiments serve the goals of the study.
The quality of the writing was improved
Author Response
Dear Reviewer,
The manuscript has been revised again. Additional background information was given and unnecessary information was deleted.
The focus of the improvements was on the introduction and the methods section. It has been made clearer why the experiments are necessary and which information adds value.
Linguistically, the entire manuscript underwent a further iteration of improvement.